# *Streptococcus pneumoniae* Causing Invasive Diseases in Children and Adults in Central Thailand, 2012–2016

**DOI:** 10.3390/vaccines10081368

**Published:** 2022-08-21

**Authors:** Wanatpreeya Phongsamart, Somporn Srifeungfung, Tanittha Chatsuwan, Pimpha Rungnobhakhun, Alan Maleesatharn, Kulkanya Chokephaibulkit

**Affiliations:** 1Department of Pediatrics, Faculty of Medicine, Siriraj Hospital, Mahidol University, Bangkok 10700, Thailand; 2Faculty of Pharmacy, Siam University, Petchkasem Road, Bangkok 10160, Thailand; 3Department of Microbiology and Antimicrobial Resistance and Center of Excellence in Antimicrobial Resistance and Stewardship, Faculty of Medicine, Chulalongkorn University, Bangkok 10330, Thailand; 4Microbiological Laboratory, Bhumipol Adulyadej Hospital, Bangkok 10220, Thailand

**Keywords:** invasive pneumococcal diseases (IPD), serotype, antimicrobial susceptibility, pneumococcal conjugate vaccine (PCV)

## Abstract

Longitudinal data regarding the serotype distribution and antimicrobial susceptibility of *S. pneumoniae*-causing invasive pneumococcal disease (IPD) in developing countries are limited. Our aim was to monitor the antimicrobial susceptibility, serotype distribution, and serotype coverage rates of the pneumococcal conjugate vaccines (PCVs) and emerging non-vaccine serotypes (NVT) between 2012 and 2016 in central Thailand. Pneumococcal isolates from sterile specimens of patients, collected within a long-standing collaborative hospital network in central Thailand between 2012 and 2016, were studied. The pneumococcal serotypes included in the 15-valent PCV were identified by the quellung reaction, while the non-PCV15 serotypes were identified by multiplex PCR. Antimicrobial susceptibilities were determined by the microbroth dilution or disk diffusion method. Of the 276 pneumococcal isolates, 129 (46.7%) were from children aged ≤5 years. Only 9.0% of patients with available data received the PCV prior to the onset of the IPD. The most common vaccine serotypes were 6B (17.4%), 19A (13.0%), and 14 (11.2%), respectively. Non-PCV15 serotypes were detected in 27.9%; the most common serotypes were 15B/C (5.1%), 15A/F (4.0%), and 23A (3.6%), respectively. The serotype coverage rates of PCV10 in children aged ≤5 years was 55.8%, and 53.3% across all ages. PCV13 provided similar coverage rates to that of PCV15, 71.3% in children aged ≤5 years, and 72.1% across all ages. High susceptibilities to cefotaxime (94.6%), ofloxacin (98.2%), linezolid (99.6%), and vancomycin (100.0%) were observed, while the susceptibility to erythromycin (50.0%), TMP-SMZ (41.3%), and tetracycline (27.2%) were low. The susceptibilities to penicillin, meropenem, and clindamycin were 85.9%, 85.9%, and 84.8%, respectively. Serotype 19A was associated with a lower susceptibility than the non-19A isolates for penicillin (75.0% vs. 87.5%, *p* = 0.045), meropenem (52.8% vs. 90.8%, *p* < 0.001), erythromycin (33.3% vs. 53.8%, *p* = 0.022), and TMP-SMZ (16.7% vs. 45.0%, *p* = 0.001). Although the majority of the pneumococcal serotypes causing IPD in central Thailand were covered by the currently available PCVs, 25% of IPD were caused by NVT. Several emerging NVT identified were 15B/C, 15A/F, and 23A. The high rates of resistance to penicillin, meropenem, erythromycin, TMP-SMZ, and tetracycline observed is a major concern. Serotype 19A was associated with lower antimicrobial susceptibilities in comparison to the non-19A serotypes.

## 1. Introduction

Pneumococcal diseases remain a leading cause of vaccine-preventable death in children less than 5 years of age. The World Health Organization (WHO) estimates that invasive pneumococcal disease (IPD) causes over one million deaths among children under five years old worldwide, annually, with the majority of these deaths occurring in developing countries. Antimicrobial resistance among *Streptococcus pneumoniae* is a global concern and, since 2007, the WHO has recommended the inclusion of pneumococcal conjugate vaccine (PCV) in national childhood immunization programs, especially in countries with high child mortality [1]. Since the introduction of the first 7-valent pneumococcal conjugate vaccine (PCV7) in 2000, global reductions in pneumococcal diseases in children under five years of age have been observed. Indeed, the number of deaths attributable to pneumococcal disease has halved, from approximately 600,000 in 2000 to 294,000 in 2015, among HIV-negative children under five years of age. It is estimated that nearly 200,000 deaths have been averted since 2000 because of PCV use [2]. However, an increase in serotypes not included in the PCV7, particularly for serotype 19A, has been observed in many countries, whether or not PCV7 was included in their national immunization program [3,4,5,6].

The introduction of higher valent PCVs—10-valent pneumococcal conjugate vaccine (PCV10) and 13-valent pneumococcal conjugate vaccine (PCV13)—has led to a further reduction in the global burden of pneumococcal diseases in children under five years of age [7,8]. Nevertheless, non-PCV13 serotypes have now emerged in many countries [9,10,11]: Hong Kong (*n* = 78), Israel (*n* = 701), Malawi (*n* = 226), South Africa (*n* = 1351), The Gambia (*n* = 203), and the USA (*n* = 674) [12].

A systematic review of the serotype distribution of *Streptococcus pneumoniae*-causing IPD in children in the post-PCV era revealed that, in countries that have introduced higher valent PCVs, the non-PCV13 serotypes contributed to 42.2% (95% CI 36.1 ± 49.5%) of the childhood IPD cases. However, regional differences were noted: 57.8% in North America, 71.9% in Europe, 45.9% in the Western Pacific, 28.5% in Latin America, and 42.7% in Africa (South Africa). The predominant non-PCV13 serotypes were 22F, 12F, 33F, 24F, 15C, 15B, 23B, 10A, and 38 [13]. It is critical to monitor the evolution of the serotype distribution and antimicrobial susceptibilities of pneumococcus-causing IPD in each setting to guide treatment and vaccine recommendations.

In Thailand, longitudinal data regarding the serotype distribution of *S. pneumoniae*-causing IPD are limited. In 2005, we initiated a collaborative network of hospital laboratories in central Thailand to collect and share pneumococcal isolates from clinical specimens. The primary objective of the network was to monitor the trend of the serotype and antimicrobial susceptibility of IPD. We have previously reported the serotype distribution of *S. pneumoniae*-causing invasive diseases in Central Thailand [14,15,16,17]. We found a significant increase of serotype 19A among children ≤5 years between 2009 and 2012 (5.6% in 2000–2009 vs. 18.3% in 2009–2012, *p* = 0.003) [16]. To our knowledge, no recent data regarding the serotype distribution and serotype coverage of PCVs in Thailand have been published since 2012.

This study aimed to build upon our earlier work by describing the serotype distribution of *S. pneumonia*-causing IPD, serotype coverage rates of PCVs, and by identifying the emerging non-vaccine serotypes in Central Thailand in 2012–2016. We also studied the antimicrobial susceptibility of pneumococcus-causing IPD. During this time period, PCV had not yet been included in the National Immunization Program (NIP) in Thailand; nevertheless, both PCV10 and PCV13 were available. The Pediatric Infectious Diseases Society of Thailand currently recommends PCV as an optional vaccine in a 3 + 1 or 2 + 1 schedule for healthy infants and children ≤5 years of age, and a 3 + 1 schedule for high-risk infants and children. Despite pediatricians generally recommending PCV use, the estimated PCV coverage among children aged ≤5 years during the study period was less than 10% countrywide and approximately 24% in central Thailand. 

A rise in drug-resistant pneumococcus is a global concern [9,10,11,12,16]. A study of the economic impact of infections attributable to drug-resistant organisms demonstrates that infections due to multidrug-resistant organisms are associated with higher mortality, longer hospital stays, and increased costs [18]. Currently, The Health Intervention and Technology Assessment Program (HITAP) in Thailand is conducting a cost-effectiveness study on the introduction of PCV into the NIP. Data regarding the serotype distribution and serotype coverage of PCVs are crucial for this analysis. In 2021, a pilot project was launched of PCV10 in a 2 + 1 schedule for children at two, four, and twelve months of age without catch-up in one of the northeastern provinces of Thailand. This underscores the importance of our current data reporting the serotypes and antimicrobial susceptibilities of IPD prior to the implementation of PCV in the NIP.

## 2. Materials and Methods

Pneumococcal isolates were collected from sterile specimens of patients in a collaborative network of 43 hospitals in Central Thailand, which included Bangkok and 15 other central provinces. The study sites included three tertiary care university hospitals in Bangkok (Siriraj Hospital, King Chulalongkorn Memorial Hospital, and Bhumibol Adulyadej Hospital), along with 40 public and private alliance hospitals. Samples collected between September 2012 and March 2016 were included in this analysis. We included all pneumococcal isolates from normal, sterile sites that grew in culture from patients of all ages being cared for in one of the 43 hospitals in our network. The pneumococcal serotypes included in the PCV15 (serotype 4, 6B, 9V, 14, 18C, 19F, 23F, 1, 5, 7F, 3, 6A, 19A, 22F and 33F) were identified by the Quellung reaction. Pneumococcal antisera from Statens Serum Institute (Copenhagen, Denmark) were used for serotyping. The calculation of the serotype coverage was performed using serotypes within the vaccine, without considering potential serogroup cross-protection. The pneumococcal isolates that were not one of the PCV15 serotypes were defined as non-vaccine types (NVT). Each NVT was identified by sequential multiplex polymerase chain reaction (PCR) using the KAPA2G fast multiplex PCR kit (KAPA Biosystems Co., Ltd., Wilmington, MA, USA), lysozyme (Sigma-Aldrich Co., Ltd., Gillingham, UK), tris-base (Affymetrix Co., Ltd., Santa Clara, CA, USA), EDTA (BIO BASIC Int., Amherst, NY, USA), and boric acid (Affymetrix Co., Ltd., Santa Clara, CA, USA) [19,20].

The antimicrobial susceptibility tests were performed according to the 2014 Clinical and Laboratory Standards Institute (CLSI) guideline [21]. The minimal inhibitory concentrations (MICs) for penicillin, cefotaxime, and meropenem were determined using the broth microdilution method. Meningitis criteria were used for penicillin and cefotaxime susceptibilities if the specimens were cerebrospinal fluid (CSF) or blood isolates from the patients diagnosed with meningitis. The susceptibilities to erythromycin, linezolid, ofloxacin, tetracycline, co-trimoxazole, and vancomycin were tested by the disk diffusion method (Antimicrobial disks, BD Diagnostics Co. Ltd., Franklin Lakes, NJ, USA). The antimicrobial susceptibilities of serotype 19A were compared to the non-19A serotypes using a Chi-square or Fisher’s exact test. The odd ratio was analyzed to indicate the effect size.

A breakthrough infection was defined as an invasive pneumococcal infection in a child who had received a ≥1 PCV10 or PCV13 dose and whose pneumococcal isolate was a PCV10 or PCV13 serotype. We defined a vaccine failure as a subset of breakthrough infections when a child had completed the age-appropriate vaccine schedule, according to the Advisory Committee on Immunization Practices (ACIP) [22] or the Pediatric Infectious Diseases Society of Thailand recommendations (i.e., 3 + 1 or 2 + 1 schedule, respectively), at least two weeks before the onset of the IPD.

## 3. Results

A total of 276 pneumococcal isolates were collected during the study period. Two hundred and sixty-three (95.3%) were blood and thirteen (4.7%) were CSF specimens. The patients’ characteristics are shown in Table 1. The age of the patients ranged from 2 months to 93 years; the most common age groups with IPD were ≤5 years (129; 46.7%) followed by ≥65 years (63; 22.8%). Two hundred and forty-nine (90.2%) isolates were from patients in the Bangkok Metropolitan Region. Among 144 patients with available clinical data, 80 (55.6%) cases had comorbidities. Of these, immunocompromised conditions were the most common underlying disease (32 cases, 42.5%), followed by heart conditions (12 cases, 15.0%), and chronic lung diseases including asthma (11 cases, 13.8%). Only 13/144 cases (9.0%) received PCV prior to the development of the IPD. The clinical outcomes from 126 patients were available and the overall case–fatality rate among these patients was 9.6%, the lowest among children aged ≤5 years (1.7%), and the highest among the elderly aged ≥65 years (17.6%).

Among 13 patients who received PCV prior to the onset of IPD, 2 completed PCV7, 2 received PCV10, and 9 received PCV13. Two children who completed PCV7 without catch-up developed IPD due to serotype 35B and 15B/C. Two children received PCV10; one child developed IPD due to serotype 23A; the other one, a four-year-old, developed serotype19A sepsis with severe necrotizing pneumonia and parapneumonic effusion, despite four doses of PCV10. Five breakthrough infections occurred in nine cases who received PCV13, but only one child had PCV13 failure, having developed serotype 14 sepsis in spite of receiving four doses of PCV13. The remaining four children who did not complete PCV13 vaccination according to their age had IPD due to serotype 19A, 14, and two cases of 6B. Another four cases who completed PCV13 vaccination according to the ACIP recommendation developed IPD from the non-PCV13 serotypes; three cases by serotype 15 A/F and the remaining by serotype 15B/C.

### 3.1. The Serotype Distribution and Coverage Rates by Pneumococcal Conjugate Vaccines

The serotype distribution of pneumococcal isolates causing IPD by different age groups is shown in Figure 1. The most common vaccine serotypes were 6B (21.7%), 14 (13.9%), and 19A (12.4%) among children ≤5 years and 6B (17.4%), 19A (13.0%), and 14 (11.2%) among all ages. Serotypes 1, 5, 22F, and 33F were not identified in our network during the study period. Non-PCV15 serotypes were found in 28.7% and 27.9% of children ≤5 years and of all ages, respectively.

The non-PCV15 serotype distribution is shown in Figure 2. The most common NVT were 15B/C (5.1%), 15A/F (4.0%), and 23A (3.6%), respectively. The serotype coverage rates of PCV10 in children ≤5 years and of all ages were 55.8% and 53.3%, respectively. PCV13 provided similar coverage rates to that of PCV15, 71.3% and 72.1%, in children ≤5 years and of all ages, respectively. The serotype coverage rate of a 23-valent pneumococcal polysaccharide vaccine (PPSV23) for patients ≥2 years was 83.7%.

### 3.2. Antimicrobial Susceptibilities of S. pneumoniae Causing IPD

The antimicrobial susceptibilities of *S. pneumoniae* isolates causing IPD is shown in Figure 3. The susceptibilities were high to cefotaxime (94.6%), ofloxacin (98.2%), linezolid (99.6%), and vancomycin (100.0%); while the susceptibilities to erythromycin (50.0%), TMP-SMZ (41.3%), and tetracycline (27.2%) were low. The susceptibilities of isolates to penicillin and meropenem were 85.9%. Of the 39 isolates that were non-susceptible to meropenem, 17 (43.5%) were serotype 19A, followed by serotype 6B (10.3%) and 23F (7.7%), respectively.

Serotype 19A was associated with lower susceptibility than the non-19A isolates to penicillin (75.0% vs. 87.5%, *p* = 0.045), meropenem (52.8% vs. 90.8%, *p* < 0.001), erythromycin (33.3% vs. 53.8, *p* = 0.022), and TMP-SMZ (16.7% vs. 45.0%, *p* = 0.001), Figure 4. PCV10 covered 29.7%, 33.3%, 28.2%, and 57.0% of the isolates that were non-susceptible to penicillin, cefotaxime, meropenem, and erythromycin, respectively. PCV13 provided higher coverage for the antimicrobial-resistant isolates, 64.9%, 80.0%, 79.5%, and 77.0% of the isolates that were non-susceptible to penicillin, cefotaxime, meropenem, and erythromycin, respectively.

Pneumococcus-causing IPD showed high susceptibilities (>90.0%) to cefotaxime, ofloxacin, linezolid, and vancomycin. The susceptibilities were low (<50.0%) for erythromycin, TMP-SMZ, and tetracycline. The susceptibilities to penicillin and meropenem were 85.9%.

Serotype 19A was significantly associated with lower susceptibility than the non-19A isolates to penicillin, meropenem, erythromycin, and TMP-SMZ.

## 4. Discussion

The majority of pneumococcus-causing IPD in central Thailand between 2012 and 2016, where PCV coverage rates among children less than 5 years of age were 7.3% countrywide and 24.0% in the Bangkok Metropolitan Region, were serotypes included in the currently available pneumococcal vaccines. The most common serotypes were 6B, 19A, and 14. This distribution of serotypes is similar to our previous report of IPD serotypes in Central Thailand between 2009 and 2012 [16], and a report of childhood IPD in Africa from 2000 to 2015, where the most common serotypes causing invasive disease (12,896 isolates) were serotypes 14, 6B, 6A, 23F, 19F, and 19A [23]. Serotype 19A was found in only 5.2% of IPD in children aged ≤5 years in our network between 2000 and 2005 and in 6.2% between 2006 and 2009 [16,17]. A significant increase in serotype 19A was observed in children aged ≤5 years (18.3%) in our network between 2009 and 2012, and it accounted for 25.0% of IPD among children ≤5 years old in 2012 [16]. In the current analysis, serotype 19A was found in 12.4% of children aged ≤5 years and was stable during the study period. Serotypes 1, 5, and 7F were rarely found to be associated with IPD in Thailand [14,15,16,24,25], which is contrary to reports from Cambodia [26] and India [27] before the introduction of PCV into the National Immunization Program. Serotypes 22F and 33F, which are included in the investigational PCV15, were not detected in our network during the study period, consistent with our previous report between 2009 and 2012 [16]. 

The serotype coverage rates of PCV10 in children aged ≤5 years and of all ages were 55.8% and 53.3%, respectively. PCV13 provided similar coverage rates to that of PCV15, 71.3% and 72.1% in children aged ≤5 years and of all ages, respectively. Thus, children should benefit from the introduction of PCVs into Thailand’s NIP, with improved serotype coverage rates for children with PCV13 or investigational PCV15 compared to PCV10. According to a recent WHO position paper, both PCV10 and PCV13 have a substantial impact against pneumonia, vaccine-type IPD, and NP carriage. Currently, there is insufficient evidence of a difference in the net impact of the two products on the overall disease burden. PCV13 may have an additional benefit in settings where the disease attributable to serotype 19A or serotype 6C is significant. The choice of the product to be implemented should be based on the programmatic characteristics, vaccine supply, vaccine price, the local and regional prevalence of vaccine serotypes, and antimicrobial resistance patterns [28]. The serotype coverage rate of a PPSV23 for patients ≥2 years was 83.7%, approximately 30%, and 10% higher compared to PCV10 and PCV13, respectively. Therefore, patients with high-risk conditions of IPD in Thailand should benefit from immunization with PPSV23, in addition to PCVs.

We observed a significant proportion of non-PCV15 serotypes causing IPD in central Thailand, 28.7% and 27.9% in children aged <5 years and of all ages, respectively. This occurred despite the low use of PCVs in children ≤5 years of age during the study period. The common non-PCV15 emerging serotypes identified in our network were 15B/C, 15A/F, and 23A, similar to studies in other countries. In German, PCV7 has been included in the national immunization program since 2006. Vaccination formulations in Germany were changed to PCV10 in April 2009 and PCV13 in December 2009. The German National Reference Center for Streptococci has conducted surveillance of IPD in children and adults since 1992. There was a significant increase in the non-PCV13 serotypes in the PCV13 period (2010–2014). Among these, the proportions of serotypes 15A and 23B increased the most. The proportion of 15A significantly increased from 0.5% during the early vaccination period (2007–2010) to 2.4% in the late vaccination period (2010–2014). The proportion of serotype 23B increased from 0.5% to 2.8% during the same period [29].

In Japan, PCV7 was licensed in 2010, followed by PCV13 in 2013. A nationwide surveillance of IPD in pediatric patients between 2015 and 2017 revealed that the most prevalent serotype was 24F, followed by 12F, 15A, and 15B/C. However, 12F increased and 24F significantly decreased during the study period (*p* < 0.001), resulting in 12F becoming the most prevalent serotype in 2017. Among the IPD isolates, the PCV7 and PCV13 coverage rates were 0.8% and 9.2%, respectively [10].

A rise of multidrug-resistant non-vaccine serotype 15A *Streptococcus pneumoniae* was also reported in the United Kingdom, where serotype 15A represented 0–4% of *S. pneumoniae* until 2008 but rose to 29% in 2013, and 32% in 2014 [30]. The emergence of NVT highlights the importance of continued surveillance to guide vaccine design and recommendations.

There were high rates of antimicrobial resistance among pneumococcus-causing IPD in central Thailand. We found that serotype 19A was associated with lower susceptibilities to various antibiotics in comparison to non-19A serotypes. The emergence of antimicrobial-resistant serotype 19A *Streptococcus pneumoniae* was reported in the US after the introduction of PCV7 and before the introduction of PCV13 [31]. Observational laboratory-based IPD surveillance data in the US using a database of Kaiser Permanente Northern California (KPNC) members with laboratory-confirmed IPD revealed that, among all ages, the proportion of *S. pneumoniae* isolates that were susceptible to penicillin, cefotaxime, and ceftriaxone generally increased during the PCV13 period. There were no reports of isolates resistant to these antimicrobials by 2018 among children aged six weeks to six years [32].

Although PCV10 covered less than 50% of the isolates that were non-susceptible to penicillin, cefotaxime, and meropenem, the majority (65–80%) of the isolates that were non-susceptible to penicillin, cefotaxime, and meropenem were serotypes included in PCV13 and the investigational PCV15. However, in our study, we did not take the cross-protection of PCV10 against serotype 19A into account. Therefore, we expect that widespread implementation of PCV in Thailand will decrease IPD caused by antimicrobial-resistant isolates, similar to those reports from other countries [33,34].

In Taiwan, ceftriaxone resistance according to non-meningitis criteria was identified in 38% of the IPD isolates from 2011 to 2013 and was a major independent risk factor associated with inappropriate initial therapy that subsequently contributed to higher mortality. The majority (77.6%) of these isolates belonged to additional PCV13 serotypes, with approximately half being 19A, 6A, or 3 [35]. Therefore, the use of PCV13 in children, as well as in the elderly population, is likely to offer protection from the infection caused by antimicrobial-resistant pneumococci. The meropenem resistance of pneumococcus-causing IPD in our network was 15.9%, and the majority were serotype 19A. Emerging meropenem resistance is expected to have a major impact on clinical practice in Thailand and, in this context, may not be the optional first-line treatment for patients. Of 629 IPD and non-IPD isolates from a nationwide surveillance in pediatric patients in Japan (2012 to 2014), the non-susceptible proportions to various antibiotics were: 46.1% to penicillin, 20.2% to cefotaxime, 20.5% to meropenem, 94.3% to erythromycin, and 0.2% to levofloxacin. The non-susceptible rate to meropenem increased, particularly for serotype 15A [36]. The high rates of resistance to meropenem are alarming.

Pneumococcal diseases significantly increase morbidity and mortality, with approximately half of these deaths occurring in children aged under five years. The consequences and deaths adversely impact individuals’ and caregivers’ work productivity. A study aimed to quantify the potential lifetime productivity loss due to pneumococcal diseases among the pediatric population in Thailand, using productivity-adjusted life years (PALYs), revealed that, in the base-case analysis, 453,401 years of life and 457,598 PALYs would be lost to pneumococcal diseases, equating to a loss of US $5586 million (95% CI 3338–10,302). Vaccination with PCV13 at birth was estimated to save 82,609 years of life and 93,759 PALYs, which equates to US $1144 million (95% CI 367–2591) in economic benefits. The disease and financial burden of pneumococcal diseases in Thailand are significant, but a large proportion of this is potentially preventable with vaccination [37]. These findings underscore the potential benefit of implementing PCVs into the NIP in Thailand and many developing countries.

Our study had several limitations. Firstly, the majority of pneumococcal isolates collected were from patients in the Bangkok Metropolitan Region. Although these data represent Central Thailand, they may not necessarily represent the whole country. Secondly, we included 276 pneumococcal isolates in the study. While this may seem a relatively small sample size, the first population-based estimates of the incidence of pneumococcal bacteremia in Southeast Asia by Baggett, et al. [25] revealed that the annual incidence of hospitalized pneumococcal bacteremia in two provinces in Thailand was 3.7 and 7.6 cases per 100,000 persons in Sa Kaeo and Nakhon Phanom, respectively. In Thailand, antibiotics are available over the counter. Data from that study revealed that, among 23,853 patients who had a blood culture performed, 7620 (32%) had received antibiotics within 72 h before the collection of the culture specimen. The rate of antibiotic use among patients aged <5 years (33%) was similar to that among patients aged ≥5 years (32%). According to these incidence data, the 276 pneumococcal isolates in our study may represent a population of 3.6 to 7.4 million. Thirdly, this is primarily a collaborative network of hospital laboratories monitoring the trend of serotype and antimicrobial susceptibility of IPD; therefore, we have limited clinical data available. Finally, the period that we collected the samples was 2012–2016. Although this is now several years ago, there have been no recently published data on the serotypes and antimicrobial susceptibilities of IPD in Thailand since our last publication, reporting between 2009–2012 [16]. In 2021, a pilot project was launched of PCV10 in a 2 + 1 schedule for children at two, four, and twelve months of age without catch-up in one of the northeastern provinces of Thailand. As such, this underscores the importance of our current data reporting serotypes and antimicrobial susceptibilities of IPD, prior to implementation of PCV in the National Immunization Program. Furthermore, IPD is not a disease under surveillance by the Ministry of Public Health of Thailand. Systemic nationwide data collection, monitoring of serotype distribution, and antimicrobial susceptibilities of pneumococcus-causing IPD are crucial to guide treatment, recommendations, and future vaccine development.

## 5. Conclusions

The majority of the pneumococcus-causing IPD in central Thailand was covered by the currently available pneumococcal vaccines. Common non-PCV15 emerging serotypes identified were 15B/C, 15A/F, and 23A. Decreased susceptibilities of pneumococcus-causing IPD to penicillin, meropenem, erythromycin, TMP-SMZ, and tetracycline are alarming and underscore the importance of increased vaccine utilization nationwide. The monitoring of IPD serotypes and antimicrobial susceptibilities is crucial to predict the vaccine efficacy, and to guide treatment recommendations and future vaccine development.

## Figures and Tables

**Figure 1 vaccines-10-01368-f001:**
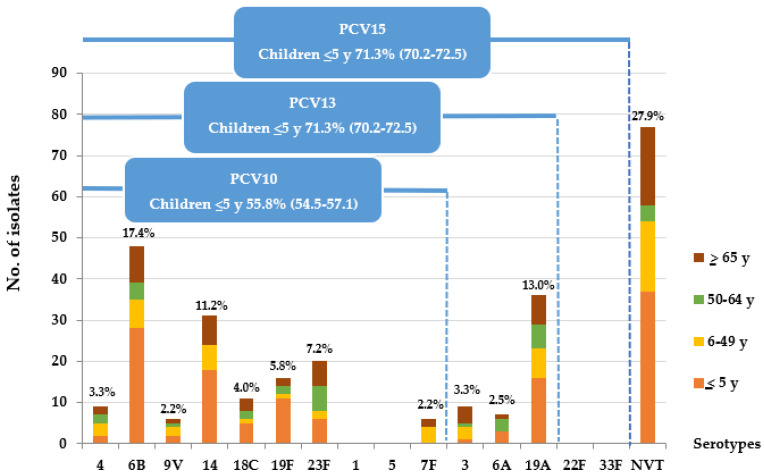
Serotype distribution of *S. pneumoniae* causing IPD, central Thailand, 2012–2016 (*n* = 276).

**Figure 2 vaccines-10-01368-f002:**
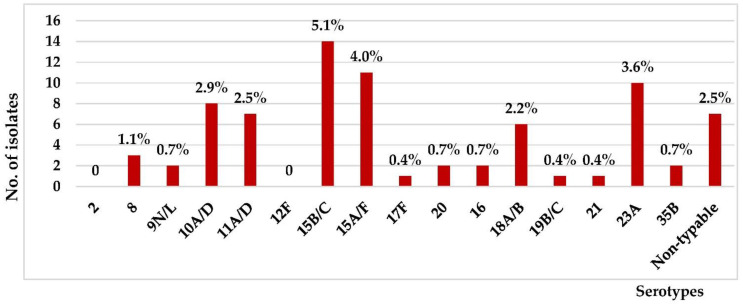
Non-PCV15 serotype distribution of *S. pneumoniae*-causing IPD, central Thailand, all ages, 2012–2016 (*n* = 77).

**Figure 3 vaccines-10-01368-f003:**
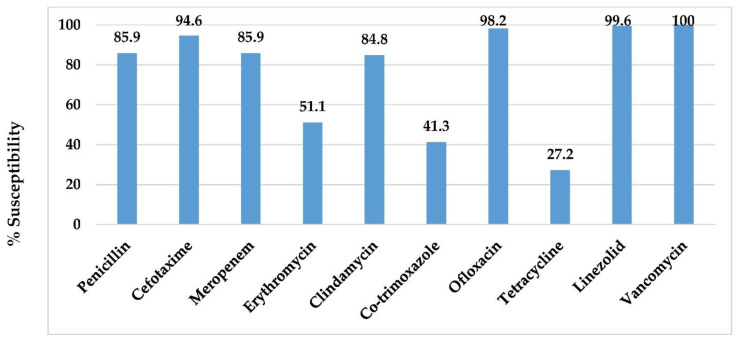
Antimicrobial susceptibilities of pneumococcus-causing IPD in central Thailand, 2012–2016.

**Figure 4 vaccines-10-01368-f004:**
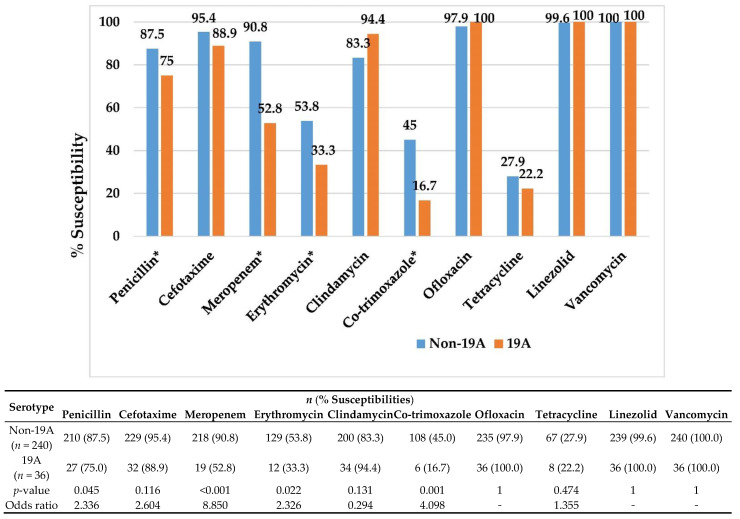
Antimicrobial susceptibilities of pneumococcus-causing IPD among serotype non-19A vs. 19A in central Thailand, 2012–2016. * statistically significant.

**Table 1 vaccines-10-01368-t001:** Characteristics of patients with invasive pneumococcal diseases with available clinical data in Central Thailand, 2012–2016.

Characteristics	*n* (%)
Age, range (*n* = 276)	2 months—93 years
≤5 years	129 (46.7)
6–49 years	53 (19.2)
50–64 years	31 (11.2)
≥65 years	63 (22.8)
Area of residence (*n* = 276)	
Bangkok and Bangkok Metropolitan	249 (90.2%)
Underlying comorbidities (*n* = 144)	
Previously healthy	64 (44.4)
Underlying comorbidities	80 (55.6)
Immunocompromised conditions	32 (42.5)
Heart condition	12 (15.0)
Chronic lung diseases including asthma	11 (13.8)
Diabetes	5 (6.3)
Asplenia	4 (5.0)
Cerebrospinal fluid leakage	2 (2.5)
History of pneumococcal vaccination prior to the development of IPD ^a^ (*n* = 144)	13 (9.0)
Outcome (*n* = 126)	
Death	12 (9.6)

^a^ IPD: invasive pneumococcal diseases.

## Data Availability

Not applicable.

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
