# Peer review of "Streptococcus pneumoniae Causing Invasive Diseases in Children and Adults in Central Thailand, 2012–2016"

_vaccines, 2022, doi:10.3390/vaccines10081368_

Round 1

Reviewer 1 Report

First of all I would like to thank for the opportunity to review this paper. The importance of studying antimicrobial susceptibility and serotype distribution of the main microorganism associated to invasive diseases is well known and emphasized due to the importance of their impact also on healthcare associated infection control. In this context, aim of the paper under review is evaluating antimicrobial susceptibility, serotype distribution, serotype coverage rates of the pneumococcal conjugate vaccines, and emerging non-vaccine serotypes between 2012 and 2016A in central Thailand.

The subject under study is certainly important, especially in the historical period we are experiencing. The article presents some interesting results, but it is nevertheless believed that, given the local impact of the same, the small sample and the different epidemiological period in which the paper was carried out (2012-2016), the manuscript is not suitable for publication. I hope that these comments will not discourage the authors and that they will take them as an impulse to improve.

Title: it can be shortened, highlight the object of the study: place, time and person. Three hospital are enrolled? Is it all the central Thailand?

Introduction: The authors should improve the introduction, making clearer what is the gap in the literature that is filled with this study. The authors must better frame their study within the vast body of literature that addressed also the issue of related costs (refer to articles with DOI: 10.2174/1389201020666190408095811) especially if the invasive infection is of nosocomial origin.

Methods: The survey was conducted using isolates from 2012-2016, I understand that the Covid-19 pandemic may have focused the researches on it, but the epidemiological data the author are presenting is old.

The enrolment procedure must be specified. How did the authors choose the way to select the samples? This can represent a great bias origin. How did they avoid the selection bias? How did the authors choose the way to select the three hospital? The authors do not propose a minimum sample size, what is the reference population? How large is it? Without the numerical identification of the reference population (all invasive infections due to S. pneumoniae) is not clear the validity of the study. A non-representative sample is by its self a non-sense-survey.

Statistical analysis: I suggest to insert a measure of the magnitude of the effect for the comparisons. Please consider to include effect sizes.

Discussion: the discussion need to be expanded. The Authors do not fully emphasize the contribution of the study to the literature. The discussion is not updated in light of the economic impact of these infections (see the above mentioned reference). The Authors should add more practical recommendations for the reader, based on their findings. Also, the section of limitations and future search is also very short, the Authors could elaborate on that.

Author Response

Response to Reviewer 1

I was invited to revise the paper entitled "Serotype Distribution and Antimicrobial Resistance of Streptococcus pneumoniae Causing Invasive Diseases in Central Thailand, 2012-2016". It was a retrospective study aimed to describe the serotype distribution of S. pneumoniae causing IPD and to identify possible emerging non-vaccine serotypes in Central Thailand. The paper focused on an important topic for public health and infectious disease prevention. Data from developing countries of Asia were poor.

Authors developed a surveillance system aimed to fill this gap. The paper is well written.

  • Thank you for your insight. Data regarding antimicrobial susceptibilities and serotype distribution are limited and poorly described in Southeast Asia.
  1. Introduction section deeply describe the study background. I suggest to add the vaccination schedule used in Thailand.
  • We agree and have added the vaccination schedule used in Thailand to the last paragraph of the introduction as follows:

“This study aimed to build upon our earlier work by describing the serotype distribution of S. pneumonia causing IPD, serotype coverage rates of PCVs, and identify the emerging non-vaccine serotypes in Central Thailand in 2012-2016. We also studied antimicrobial susceptibility of pneumococcus causing IPD. During this time period PCV had not yet been included in the National immunization program (NIP) in Thailand, nevertheless, both PCV10 and PCV13 were available. The Pediatric Infectious Diseases Society of Thailand currently recommends PCV as an optional vaccine in a 3+1 or 2+1 schedule for healthy infants and children < 5 years of age, and also a 3+1 schedule for high-risk infants and children. Although pediatricians have been generally recommending PCV vaccination, the estimated PCV coverage among children ages <5 years during the study period was less than 10% countrywide and approximately 24% in central Thailand.”

  1. Methods are adequate.
  • Acknowledged.
  1. Results should be presented clearly. I suggest to add a table about patient’s characteristics. It can help the readability of this section.
  • We agree and have added a new table summarizing the patients’ characteristics in the Results section:

Table 1: Characteristics of patients with invasive pneumococcal diseases with available clinical data in Central Thailand, 2012-2016

Characteristics

N (%)

Age, range (n =276)

   < 5 years

   6-49 years

   50-64 years

   > 65 years

2 months – 93 years

129 (46.7)

53 (19.2)

31 (11.2)

63 (22.8)

Area of residence (n=276)

   Bangkok and Bangkok metropolitan

249 (90.2%)

Underlying comorbidities (n =144)

   Previously healthy

   Underlying comorbidities

      Immunocompromised conditions

      Heart condition

      Chronic lung diseases including asthma

      Diabetes

      Asplenia

      Cerebrospinal fluid leakage

64 (44.4)

80 (55.6)

32 (42.5)

12 (15.0)

11 (13.8)

5 (6.3)

4 (5.0)

2 (2.5)

History of pneumococcal vaccination prior to development of IPDa (n=144)

13 (9.0)

Outcome (n =126)

    Death

12 (9.6)

aIPD: invasive pneumococcal diseases

  1. In addition, if available, Authors should add the outcomes of these cases.
  • Data on outcomes were available from 126 patients, 12 patients died giving a case-fatality rates of 9.6%. These data are included in the last sentence of the first paragraph of the results section and also in Table 1.
  1. Other minor observation: some lines used a wrong font (example 1-2 or 207 or 228 or 274).
  • We have corrected the front to be consistent throughout the manuscript.

Reviewer 2 Report

I was invited to revise the paper entitled "Serotype Distribution and Antimicrobial Resistance of Streptococcus pneumoniae Causing Invasive Diseases in Central Thailand, 2012-2016". It was a retrospective study aimed to describe the serotype distribution of S. pneumonia causing IPD and to identify possible emerging non-vaccine serotypes in Central Thailand. The paper focused on a important topic for public health and infectious disease prevention. Data from developing countries of Asia were poor.

Authors developed a surveillnce system aimed to fill this gap.

The paper is well written. Introduction section deeply describe the study background. I suggest to add the vaccination schedule used in Thailand.

Methods are adequate.

Results should be presented clearly. I suggest to add a table about patients characteristics. It can help the readability of this section. In addition, if available, Authors should add the outcomes of these cases.

Other minor observation: some lines used a wrong font (exaple 1-2 or 207 or 228 or 274).

Author Response

I was invited to revise the paper entitled "Serotype Distribution and Antimicrobial Resistance of Streptococcus pneumoniae Causing Invasive Diseases in Central Thailand, 2012-2016". It was a retrospective study aimed to describe the serotype distribution of S. pneumoniae causing IPD and to identify possible emerging non-vaccine serotypes in Central Thailand. The paper focused on an important topic for public health and infectious disease prevention. Data from developing countries of Asia were poor.

Authors developed a surveillance system aimed to fill this gap. The paper is well written.

  • Thank you for your insight. Data regarding antimicrobial susceptibilities and serotype distribution are limited and poorly described in Southeast Asia.
  1. Introduction section deeply describe the study background. I suggest to add the vaccination schedule used in Thailand.
  • We agree and have added the vaccination schedule used in Thailand to the last paragraph of the introduction as follows:

“This study aimed to build upon our earlier work by describing the serotype distribution of S. pneumonia causing IPD, serotype coverage rates of PCVs, and identify the emerging non-vaccine serotypes in Central Thailand in 2012-2016. We also studied antimicrobial susceptibility of pneumococcus causing IPD. During this time period PCV had not yet been included in the National immunization program (NIP) in Thailand, nevertheless, both PCV10 and PCV13 were available. The Pediatric Infectious Diseases Society of Thailand currently recommends PCV as an optional vaccine in a 3+1 or 2+1 schedule for healthy infants and children < 5 years of age, and also a 3+1 schedule for high-risk infants and children. Although pediatricians have been generally recommending PCV vaccination, the estimated PCV coverage among children ages <5 years during the study period was less than 10% countrywide and approximately 24% in central Thailand.”

  1. Methods are adequate.
  • Acknowledged.
  1. Results should be presented clearly. I suggest to add a table about patient’s characteristics. It can help the readability of this section.
  • We agree and have added a new table summarizing the patients’ characteristics in the Results section:

Table 1: Characteristics of patients with invasive pneumococcal diseases with available clinical data in Central Thailand, 2012-2016

Characteristics

N (%)

Age, range (n =276)

< 5 years

6-49 years

50-64 years

> 65 years

2 months – 93 years

129 (46.7)

53 (19.2)

31 (11.2)

63 (22.8)

Area of residence (n=276)

Bangkok and Bangkok metropolitan

249 (90.2%)

Underlying comorbidities (n =144)

Previously healthy

Underlying comorbidities

Immunocompromised conditions

Heart condition

Chronic lung diseases including asthma

Diabetes

Asplenia

Cerebrospinal fluid leakage

64 (44.4)

80 (55.6)

32 (42.5)

12 (15.0)

11 (13.8)

5 (6.3)

4 (5.0)

2 (2.5)

History of pneumococcal vaccination prior to development of IPDa (n=144)

13 (9.0)

Outcome (n =126)

Death

12 (9.6)

aIPD: invasive pneumococcal diseases

  1. In addition, if available, Authors should add the outcomes of these cases.
  • Data on outcomes were available from 126 patients, 12 patients died giving a case-fatality rates of 9.6%. These data are included in the last sentence of the first paragraph of the results section and also in Table 1.
  1. Other minor observation: some lines used a wrong font (example 1-2 or 207 or 228 or 274).
  • We have corrected the front to be consistent throughout the manuscript.

Round 2

Reviewer 1 Report

The subject under study is important and the article presents some interesting results, but given the local impact of the same, the small sample and the different epidemiological period in which the paper was carried out (2012-2016), the manuscript is not suitable for publication at this stage.

I did precise comments, if the Authors will consider to provide a point by point changes to the manuscript I can consider further, but at this stage the decision is confirmed. Following my previous revision suggestions.  

Titleit can be shortened, highlight the object of the study: place, time and person. Three hospital are enrolled? Is it all the central Thailand?

IntroductionThe authors should improve the introduction, making clearer what is the gap in the literature that is filled with this study. The authors must better frame their study within the vast body of literature that addressed also the issue of related costs (refer to articles with DOI: 10.2174/1389201020666190408095811) especially if the invasive infection is of nosocomial origin.

Methods: The survey was conducted using isolates from 2012-2016, I understand that the Covid-19 pandemic may have focused the researches on it, but the epidemiological data the author are presenting is old.

The enrolment procedure must be specified. How did the authors choose the way to select the samples? This can represent a great bias origin. How did they avoid the selection bias? How did the authors choose the way to select the three hospital? The authors do not propose a minimum sample size, what is the reference population? How large is it? Without the numerical identification of the reference population (all invasive infections due to S. pneumoniae) is not clear the validity of the study. A non-representative sample is by its self a non-sense-survey.

Statistical analysis: I suggest to insert a measure of the magnitude of the effect for the comparisons. Please consider to include effect sizes.

Discussion: the discussion need to be expanded. The Authors do not fully emphasize the contribution of the study to the literature. The discussion is not updated in light of the economic impact of these infections (see the above mentioned reference). The Authors should add more practical recommendations for the reader, based on their findings. Also, the section of limitations and future search is also very short, the Authors could elaborate on that.

Author Response

Response to Reviewer 1

First of all I would like to thank for the opportunity to review this paper. The importance of studying antimicrobial susceptibility and serotype distribution of the main microorganism associated to invasive diseases is well known and emphasized due to the importance of their impact also on healthcare associated infection control. In this context, aim of the paper under review is evaluating antimicrobial susceptibility, serotype distribution, serotype coverage rates of the pneumococcal conjugate vaccines, and emerging non-vaccine serotypes between 2012 and 2016 in central Thailand.

The subject under study is certainly important, especially in the historical period we are experiencing. The article presents some interesting results, but it is nevertheless believed that, given the local impact of the same, the small sample and the different epidemiological period in which the paper was carried out (2012-2016), the manuscript is not suitable for publication. I hope that these comments will not discourage the authors and that they will take them as an impulse to improve.

  • Thank you for your constructive comments. We would like to highlight that while the epidemiological period report was 2012-2016 there has been no other published data on serotypes and antimicrobial susceptibilities of IPD in Thailand since our prior publication which covered 2009-2012 [Hum Vaccin Immunother. 2014;10(7):1866-73]. We strongly believe that these data are important for health care workers in the region and should be published. Also, this data collection is ongoing and further epidemiological studies over the last few years will be possible but this has been challenging in the context of the global COVID-19 pandemic. In 2021, a pilot project was launched of PCV-10 in a 2+1 schedule for children at 2, 4, and 12 months of age without catch up in one of the Northeastern provinces of Thailand. As such, this underscores the importance of our current data reporting serotypes and antimicrobial susceptibilities of IPD prior to implementation of PCV in the National Immunization Program. We have added in the discussion that the epidemiological period reported is a limitation but also emphasize the lack of other data during this time period.

  1. Titleit can be shortened, highlight the object of the study: place, time and person. Three hospital are enrolled? Is it all the central Thailand?

  • Pneumococcal isolates from sterile specimens of patients were collected in a collaborative network of 43 hospitals in Central Thailand (which included Bangkok and 15 other central provinces). The study sites included 3 tertiary care University hospitals in Bangkok [Siriraj Hospital (the largest University Hospital in Thailand), King Chulalongkorn Memorial Hospital (a large University Hospital in Thailand), and Bhumibol Adulyadej Hospital (a University and a military Hospital)], along with 40 public and private alliance hospitals.

We have modified the title to: “Serotype Distribution and Antimicrobial Resistance of Streptococcus pneumoniae Causing Invasive Diseases in Children and Adults in Central Thailand, 2012-2016”, which now highlights the objects of the study including the study population, location and period of time.

  1. IntroductionThe authors should improve the introduction, making clearer what is the gap in the literature that is filled with this study. The authors must better frame their study within the vast body of literature that addressed also the issue of related costs (refer to articles with DOI: 10.2174/1389201020666190408095811) especially if the invasive infection is of nosocomial origin.

.

  • We have included in the introduction current knowledge gaps and also highlighted the importance of the serotype distribution and antimicrobial resistance to cost-effectiveness analyses which guide national policies. We did not emphasize the costs related to nosocomial acquired infection as most invasive pneumococcal diseases originated from nasopharyngeal colonization. The Introduction has been revised as follows:

“A systematic review of serotype distribution of Streptococcus pneumoniae causing IPD in children in the post-PCV era revealed that in countries that have introduced higher valent PCVs, the non-PCV13 serotypes contributed to 42.2% (95%CI 36.1+49.5%) of childhood IPD cases. However, regional differences were noted, 57.8% in North America, 71.9% in Europe, 45.9% in the Western Pacific, 28.5% in Latin America, and 42.7% in an African country (South Africa). Predominant non-PCV13 serotypes were 22F, 12F, 33F, 24F, 15C, 15B, 23B, 10A, and 38 [13]. It is critical to monitor the evolution of the serotype distribution and antimicrobial susceptibilities of pneumococcus causing IPD in each setting to guide treatment and vaccine recommendations

In Thailand, longitudinal data regarding serotype distribution of S. pneumoniae causing IPD are limited. In 2005, we initiated a collaborative network of hospital laboratories in central Thailand to collect and share pneumococcal isolates from clinical specimens. The primary objective of the network was to monitor the trend of serotype and antimicrobial susceptibility of IPD. We have previously reported the serotype distribution of S. pneumoniae causing invasive diseases in Central Thailand [14-17]. We found a significant increase of serotype 19A among children ≤5 years between 2009 and 2012 (5.6% in 2000-2009 vs 18.3% in 2009-2012, P = 0.003) [16]. To our knowledge, no recent data regarding serotype distribution and serotype coverage of PCVs in Thailand have been published after 2012.

This study aimed to build upon our earlier work by describing the serotype distribution of S. pneumonia causing IPD, serotype coverage rates of PCVs, and identify the emerging non-vaccine serotypes in Central Thailand in 2012-2016. We also studied antimicrobial susceptibility of pneumococcus causing IPD. During this time period PCV had not yet been included in the National immunization program (NIP) in Thailand, nevertheless, both PCV10 and PCV13 were available. The Pediatric Infectious Diseases Society of Thailand recommended PCV as an optional vaccine in a 3+1 or 2+1 schedule for children < 5 years of age and also a 3+1 schedule for high-risk children. Despite pediatricians generally recommending PCV use the estimated PCV coverage among children ages <5 years during the study period was less than 10% countrywide and approximately 30% in central Thailand.

Currently, The Health Intervention and Technology Assessment Program (HITAP) in Thailand is conducting a cost-effectiveness study of introduction PCV into the NIP. Data regarding serotype distribution, serotype coverage of PCVs are crucial for this analysis for the country.  In 2021, a pilot project was launched of PCV-10 in a 2+1 schedule for children at 2, 4, and 12 months of age without catch up in one of the Northeastern provinces of Thailand. This underscores the importance of our current data reporting serotypes and antimicrobial susceptibilities of IPD prior to implementation of PCV in the NIP. “

  1. Methods: 
    • The survey was conducted using isolates from 2012-2016, I understand that the Covid-19 pandemic may have focused the researches on it, but the epidemiological data the author are presenting is old.

  • We would like to highlight that while the epidemiological period report was 2012-2016 there has been no other published data on serotypes and antimicrobial susceptibilities of IPD in Thailand since our prior publication which covered 2009-2012 [Hum Vaccin Immunother. 2014;10(7):1866-73]. We strongly believe that these data are important for health care professional in the region and should be published.

  • The enrolment procedure must be specified. How did the authors choose the way to select the samples? This can represent a great bias origin. How did they avoid the selection bias? How did the authors choose the way to select the three hospital?

  • We included all culture positive samples from sterile specimens isolated from patients cared for in one of the 43 hospitals in our network. While selection bias is always a concern we believe there is minimal selective bias in our sample selection. We have added the enrollment procedure and described our clinical site network in the first paragraph of the materials and methods as follows:

“Pneumococcal isolates from sterile specimens of patients in a collaborative network of 43 hospitals in Central Thailand, which included Bangkok and 15 other central provinces. The study sites included 3 tertiary care University hospitals in Bangkok  (Siriraj Hospital, King Chulalongkorn Memorial Hospital, and Bhumibol Adulyadej Hospital), and 40 public and private alliance hospitals . Samples collected between September 2012 and March 2016 were included in this analysis. We included all pneumococcal isolates from normal sterile sites that grew in culture from patients of all ages being cared for in one of the 43 hospitals in our network. Pneumococcal serotypes included in the PCV15 (serotype 4, 6B, 9V, 14, 18C, 19F, 23F, 1, 5, 7F, 3, 6A, 19A, 22F and 33F) were identified by Quellung reaction. Pneumococcal antisera from Statens Serum Institute (Copenhagen, Denmark) were used for serotyping. Calculation of serotype coverage was performed using serotypes within the vaccine, without considering potential serogroup cross-protection. Pneumococcal isolates that were not one of the PCV15 serotypes were defined as non-vaccine types (NVT). Each NVT was identified by sequential multiplex polymerase chain reaction (PCR) using the KAPA2G fast multiplex PCR kit (KAPA Biosystems Co., Ltd., USA), lysozyme (Sigma-Aldrich Co., Ltd., United Kingdom), tris-base (Affymetrix Co., Ltd., USA), EDTA (BIO BASIC Int., USA) and boric acid (Affymetrix Co., Ltd., USA) [18-19].”

  • The authors do not propose a minimum sample size, what is the reference population? How large is it? Without the numerical identification of the reference population (all invasive infections due to S. pneumoniae) is not clear the validity of the study. A non-representative sample is by its self a non-sense-survey.

  • We did not propose a minimum sample size but included all positive culture samples (as described in our responses above). Data regarding the incidence of invasive pneumococcal disease (IPD) in Thailand and other countries in Southeast Asia are limited. Baggett, et al reported the first population-based estimates of the incidence of pneumococcal bacteremia in Southeast Asia. The annual incidence of hospitalized pneumococcal bacteremia in 2 provinces in Thailand was 3.7 and 7.6 cases per 100,000 persons in Sa Kaeo and Nakhon Phanom, respectively. In Thailand, antibiotics are available over the counter. Data from that study revealed that among 23,853 patients who had blood culture performed, 7620 (32%) had received antibiotics within 72 hours before collection of the culture specimen. The rate of antibiotic use among patients aged <5 years (33%) was similar to that among patients aged >5 years (32%). According to these incidence data, 276 pneumococcal isolates may represent a population of 3.6 to 7.4 million. We have added the small sample size as a limitation in the discussion as follows

“Our study had several limitations. Firstly, the majority of pneumococcal isolates collected were from patients in the Bangkok Metropolitan area. Although these data represent Central Thailand it may not necessarily represent the whole country. Secondly, we included 276 pneumococcal isolates in the study and while this may seem a relatively small sample size the first population-based estimates of the incidence of pneumococcal bacteremia in Southeast Asia by Baggett, et al (24) revealed that the annual incidence of hospitalized pneumococcal bacteremia in was 3.7 and 7.6 cases per 100,000 persons in Sa Kaeo and Nakhon Phanom provinces in Thailand, respectively. According to these incidence data, the 276 pneumococcal isolates in our study may represent a population of 3.6 to 7.4 million. Thirdly, this is a primarily a collaborative network of hospital laboratories to monitor the trend of serotype and antimicrobial susceptibility of IPD, therefore we have limited clinical data available. Finally, the period that we collected the samples was 2012-2016. Although this is now several years ago there has been no recent published data on serotypes and antimicrobial susceptibilities of IPD in Thailand since our last publication reporting between 2009-2012 (16). In 2021, a pilot project was launched of PCV-10 in a 2+1 schedule for children at 2, 4, and 12 months of age without catch up in one of the Northeastern provinces of Thailand. As such, this underscores the importance of our current data reporting serotypes and antimicrobial susceptibilities of IPD prior to implementation of PCV in the National Immunization Program. Furthermore, IPD is not a disease under surveillance by the Ministry of Public Health of Thailand.  Systemic Nationwide data collection, monitoring of serotype distribution and antimicrobial susceptibilities of pneumococcus causing IPD is crucial to guide treatment recommendation and future vaccine development.”

  1. Statistical analysis: I suggest to insert a measure of the magnitude of the effect for the comparisons. Please consider to include effect sizes.

  • We have added a table that include the odds ratio into Figure 4 as suggested.

Serotype

N (% Susceptibilities)

Penicillin

Cefotaxime

Meropenem

Erythromycin

Clindamycin

Co-trimoxazole

Ofloxacin

Tetracycline

Linezolid

Vancomycin

Non-19A

(n =240)

210 (87.5)

229 (95.4)

218 (90.8)

129 (53.8)

200 (83.3)

108 (45.0)

235 (97.9)

67 (27.9)

239 (99.6)

240 (100.0)

19A

(n =36)

27 (75.0)

32 (88.9)

19 (52.8)

12 (33.3)

34 (94.4)

6 (16.7)

36 (100.0)

8 (22.2)

36 (100.0)

36 (100.0)

p-value

0.045

0.116

<0.001

0.022

0.131

0.001

1

0.474

1

1

Odds ratio

2.336

2.604

8.850

2.326

0.294

4.098

-

1.355

-

-

Figure 4. Antimicrobial susceptibilities of pneumococcus causing IPD among serotype non-19A VS 19A in central Thailand, 2012-2016.

  1. Discussion:
    • the discussion need to be expanded. The Authors do not fully emphasize the contribution of the study to the literature.
  • We have added the contribution of this study in the discussion as follows

5.1.1 Serotype distribution and serotype coverage by vaccines in the 2nd paragraph of the discussion:

“The serotype coverage rates of PCV10 in children <5 years and all ages were 55.8% and 53.3%, respectively. PCV13 provided similar coverage rates to that of PCV15, 71.3% and 72.1% in children <5 years and all ages, respectively. Thus, children should benefit from introduction of PCVs into Thailand’s NIP, with improved serotype coverage rates for children with PCV13 or investigational PCV15 compared to PCV10. According to a recent WHO position paper, both PCV10 and PCV13 have a substantial impact against pneumonia, vaccine-type IPD and NP carriage. Currently, there is insufficient evidence of a difference in the net impact of the 2 products on overall disease burden. PCV13 may have an additional benefit in settings where disease attributable to serotype 19A or serotype 6C is significant. The choice of product to be implemented should be based on programmatic characteristics, vaccine supply, vaccine price, the local and regional prevalence of vaccine serotypes and antimicrobial resistance patterns [27]. The serotype coverage rate of a PPSV23 for patients >2 years was 83.7%, approximately 30% and 10% higher compared to PCV10 and PCV13, respectively. Therefore, patients with high risk conditions of IPD in Thailand should benefit from immunization with PPSV23, in addition to PCVs.”

  1. World Health Organization. Pneumococcal conjugate vaccines in infants and children under 5 years of age: WHO position paper. Wkly Epidemiol Rec 2019, 94, 85–104.

5.1.2 Antimicrobial resistance

  • We emphasize the high rates of meropenem resistance of pneumococcus causing IPD in our network and to be cautious using meropenem as a first-line treatment.

“In Taiwan, ceftriaxone resistance according to non-meningitis criteria was identified in 38% of the IPD isolates from 2011 to 2013, and a major independent risk factor associated with inappropriate initial therapy that subsequently contributed to higher mortality. The majority (77.6%) of these isolates belonged to additional PCV13 serotypes, with approximately half being19A, 6A or 3 [33]. Therefore, the use of PCV13 in children as well as in the elderly population is likely to offer protection from the infection caused by antimicrobial-resistant pneumococci. Meropenem resistance of pneumococcus causing IPD in our network was 15.9%, and the majority were serotype 19A. Emerging meropenem resistance is expected to have a major impact on clinical practice in Thailand and in this context may not be the optional first-line treatment for patients. Of 629 IPD and non-IPD isolates from a nationwide surveillance in pediatric patients in Japan (2012 to 2014), the non-susceptible proportions to various antibiotics were: 46.1% to penicillin, 20.2% to cefotaxime, 20.5% to meropenem, 94.3% to erythromycin and 0.2% to levofloxacin. The non-susceptible rate to meropenem increased, particularly for serotype 15A [34]. High rates of resistance to meropenem is alarming.“

  1. The discussion is not updated in light of the economic impact of these infections (see the above mentioned reference).

  • We have added the economic impact of pneumococcal diseases in the Discussion.

Pneumococcal diseases significantly increase morbidity and mortality, with approximately half of these death occurring in children aged under 5 years. The consequences and deaths adversely impact individuals’ and caregivers’ work productivity. A study aimed to quantify the potential lifetime productivity loss due to pneumococcal diseases among the pediatric population in Thailand using productivity-adjusted life years (PALYs) revealed that 453,401 years of life and 457,598 PALYs would be lost to pneumococcal diseases, equating to a loss of US$5586 (95% CI 3338–10,302) million. Vaccination with PCV13 at birth was estimated to save 82,609 years of life and 93,759 PALYs, which equated to US$1144 (95% CI 367–2591) million in economic benefits. The disease and financial burden of pneumococcal diseases in Thailand is significant, but a large proportion of this is potentially preventable with vaccination [36]. These findings underscore the potential benefit of implementation of PCVs in to the NIP in Thailand and many developing countries.

  1. Ounsirithupsakul, T.; Dilokthornsakul, P.; Kongpakwattana, K.; et al. Esti-mating the Productivity Burden of Pediatric Pneumococcal Disease in Thailand. Appl Health Econ Health Policy

2020, 18, 579-587.

  1. The Authors should add more practical recommendations for the reader, based on their findings.

  • We added information regarding choice of PCVs and recommended not to use meropenem as an initial treatment of suspected IPD due to high rates of resistance as mention above.

  1. Also, the section of limitations and future search is also very short, the Authors could elaborate on that.
  • We have added a paragraph highlighting the study limitations in the discussion as follows:

“Our study had several limitations. Firstly, the majority of pneumococcal isolates collected were from patients in the Bangkok Metropolitan area. Although these data represent Central Thailand it may not necessarily represent the whole country. Secondly, we included 276 pneumococcal isolates in the study and while this may seem a relatively small sample size the first population-based estimates of the incidence of pneumococcal bacteremia in Southeast Asia by Baggett, et al (24) revealed that the annual incidence of hospitalized pneumococcal bacteremia in was 3.7 and 7.6 cases per 100,000 persons in Sa Kaeo and Nakhon Phanom provinces in Thailand, respectively. According to these incidence data, the 276 pneumococcal isolates in our study may represent a population of 3.6 to 7.4 million. Thirdly, this is a primarily a collaborative network of hospital laboratories to monitor the trend of serotype and antimicrobial susceptibility of IPD, therefore we have limited clinical data available. Finally, the period that we collected the samples was 2012-2016. Although this is now several years ago there has been no recent published data on serotypes and antimicrobial susceptibilities of IPD in Thailand since our last publication reporting between 2009-2012 (16). In 2021, a pilot project was launched of PCV-10 in a 2+1 schedule for children at 2, 4, and 12 months of age without catch up in one of the Northeastern provinces of Thailand. As such, this underscores the importance of our current data reporting serotypes and antimicrobial susceptibilities of IPD prior to implementation of PCV in the National Immunization Program. Furthermore, IPD is not a disease under surveillance by the Ministry of Public Health of Thailand.  Systemic Nationwide data collection, monitoring of serotype distribution and antimicrobial susceptibilities of pneumococcus causing IPD is crucial to guide treatment recommendation and future vaccine development.”

Round 3

Reviewer 1 Report

The problem is not to reply but how the Author modified the paper. I did precise comments, if the Authors will consider providing point by point changes to the manuscript I can consider further, but at this stage the decision is confirmed. Following my previous revision suggestions please.  

The subject under study is important and the article presents some interesting results, but given the local impact of the same, the small sample and the different epidemiological period in which the paper was carried out (2012-2016), the manuscript is not suitable for publication at this stage.

Titleit can be shortened, highlight the object of the study: place, time and person. Three hospital are enrolled? Is it all the central Thailand?

IntroductionThe authors should improve the introduction, making clearer what is the gap in the literature that is filled with this study. The authors must better frame their study within the vast body of literature that addressed also the issue of related costs (refer to articles with DOI: 10.2174/1389201020666190408095811) especially if the invasive infection is of nosocomial origin.

Methods: The survey was conducted using isolates from 2012-2016, I understand that the Covid-19 pandemic may have focused the researches on it, but the epidemiological data the author are presenting is old.

The enrolment procedure must be specified. How did the authors choose the way to select the samples? This can represent a great bias origin. How did they avoid the selection bias? How did the authors choose the way to select the three hospital? The authors do not propose a minimum sample size, what is the reference population? How large is it? Without the numerical identification of the reference population (all invasive infections due to S. pneumoniae) is not clear the validity of the study. A non-representative sample is by its self a non-sense-survey.

Discussion: the discussion need to be expanded. The Authors do not fully emphasize the contribution of the study to the literature. The discussion is not updated in light of the economic impact of these infections (see the above mentioned reference). The Authors should add more practical recommendations for the reader, based on their findings. Also, the section of limitations and future search is also very short, the Authors could elaborate on that.

Author Response

10 August 2022

Vaccines

RE: Manuscript ID vaccines-1812705

Article title: Serotype Distribution and Antimicrobial Resistance of Streptococcus pneumoniae Causing Invasive Diseases in Central Thailand, 2012-2016

Thank you for providing the reviewers’ comments and inviting us to resubmit a revised version of our manuscript. Please find enclosed the revised manuscript to be considered for publication along with a point-by-point response to the reviewers’ comments. We leave all the previous revision (in yellow) and put the additional revision (in blue) in response further to the reviewer’s comments (reviewer#1, point 2, 3, 7) that we might not addressed adequately in the previous revision.

This article has never been published elsewhere or submitted simultaneously for publication. All authors have contributed to, seen, and approved the final, submitted version of the manuscript.

Kulkanya Chokephaibulkit, MD

Professor of Pediatrics,

Department of Pediatrics and Siriraj Institute of Clinical Research,

Faculty of Medicine Siriraj Hospital, Mahidol University, Bangkok, Thailand

Wanglang road

Bangkok, 10700

Thailand

Telephone: +66-241-80544

Email:  kulkanya.cho@mahidol.ac.th

Response to Editorial Comments to Author:

Response to Reviewer 1

First of all I would like to thank for the opportunity to review this paper. The importance of studying antimicrobial susceptibility and serotype distribution of the main microorganism associated to invasive diseases is well known and emphasized due to the importance of their impact also on healthcare associated infection control. In this context, aim of the paper under review is evaluating antimicrobial susceptibility, serotype distribution, serotype coverage rates of the pneumococcal conjugate vaccines, and emerging non-vaccine serotypes between 2012 and 2016 in central Thailand.

The subject under study is certainly important, especially in the historical period we are experiencing. The article presents some interesting results, but it is nevertheless believed that, given the local impact of the same, the small sample and the different epidemiological period in which the paper was carried out (2012-2016), the manuscript is not suitable for publication. I hope that these comments will not discourage the authors and that they will take them as an impulse to improve.

  • Thank you for your constructive comments. We would like to highlight that while the epidemiological period report was 2012-2016 there has been no other published data on serotypes and antimicrobial susceptibilities of IPD in Thailand since our prior publication which covered 2009-2012 [Hum Vaccin Immunother. 2014;10(7):1866-73]. We strongly believe that these data are important for health care workers in the region and should be published. Also, this data collection is ongoing and further epidemiological studies over the last few years will be possible but this has been challenging in the context of the global COVID-19 pandemic. In 2021, a pilot project was launched of PCV-10 in a 2+1 schedule for children at 2, 4, and 12 months of age without catch up in one of the Northeastern provinces of Thailand. As such, this underscores the importance of our current data reporting serotypes and antimicrobial susceptibilities of IPD prior to implementation of PCV in the National Immunization Program. We have added in the discussion that the epidemiological period reported is a limitation but also emphasize the lack of other data during this time period.

Title

Point 1: it can be shortened, highlight the object of the study: place, time and person.

Response 1:

  • We have modified the title to: “Streptococcus pneumoniae Causing Invasive Diseases in Children and Adults in Central Thailand, 2012-2016”, which now shortened, and highlights the objects of the study including the study population, location and period of time.

Point 2: Three hospital are enrolled? Is it all the central Thailand?

Response 2:

  • Pneumococcal isolates from sterile specimens of patients were collected in a collaborative network of 43 hospitals in Central Thailand (which included Bangkok and 15 other central provinces). The study sites included 3 tertiary care University hospitals in Bangkok [Siriraj Hospital (the largest University Hospital in Thailand), King Chulalongkorn Memorial Hospital (a large University Hospital in Thailand), and Bhumibol Adulyadej Hospital (a University and a military Hospital)], along with 40 public and private alliance hospitals. Therefore, it represents the central Thailand.

Introduction

Point 3. The authors should improve the introduction, making clearer what is the gap in the literature that is filled with this study. The authors must better frame their study within the vast body of literature that addressed also the issue of related costs (refer to articles with DOI: 10.2174/1389201020666190408095811) especially if the invasive infection is of nosocomial origin.

Response 3:

  • We have included in the introduction current knowledge gaps and also highlighted the importance of the serotype distribution and antimicrobial resistance to cost-effectiveness analyses which guide national policies. We also have added the issue of economic impact related to drug-resistant infections and reference as suggested. The Introduction has been revised as follows:

“A systematic review of serotype distribution of Streptococcus pneumoniae causing IPD in children in the post-PCV era revealed that in countries that have introduced higher valent PCVs, the non-PCV13 serotypes contributed to 42.2% (95%CI 36.1+49.5%) of childhood IPD cases. However, regional differences were noted, 57.8% in North America, 71.9% in Europe, 45.9% in the Western Pacific, 28.5% in Latin America, and 42.7% in an African country (South Africa). Predominant non-PCV13 serotypes were 22F, 12F, 33F, 24F, 15C, 15B, 23B, 10A, and 38 [13]. It is critical to monitor the evolution of the serotype distribution and antimicrobial susceptibilities of pneumococcus causing IPD in each setting to guide treatment and vaccine recommendations

In Thailand, longitudinal data regarding serotype distribution of S. pneumoniae causing IPD are limited. In 2005, we initiated a collaborative network of hospital laboratories in central Thailand to collect and share pneumococcal isolates from clinical specimens. The primary objective of the network was to monitor the trend of serotype and antimicrobial susceptibility of IPD. We have previously reported the serotype distribution of S. pneumoniae causing invasive diseases in Central Thailand [14-17]. We found a significant increase of serotype 19A among children ≤5 years between 2009 and 2012 (5.6% in 2000-2009 vs 18.3% in 2009-2012, P = 0.003) [16]. To our knowledge, no recent data regarding serotype distribution and serotype coverage of PCVs in Thailand have been published after 2012.

This study aimed to build upon our earlier work by describing the serotype distribution of S. pneumonia causing IPD, serotype coverage rates of PCVs, and identify the emerging non-vaccine serotypes in Central Thailand in 2012-2016. We also studied antimicrobial susceptibility of pneumococcus causing IPD. During this time period PCV had not yet been included in the National immunization program (NIP) in Thailand, nevertheless, both PCV10 and PCV13 were available. The Pediatric Infectious Diseases Society of Thailand recommended PCV as an optional vaccine in a 3+1 or 2+1 schedule for children < 5 years of age and also a 3+1 schedule for high-risk children. Despite pediatricians generally recommending PCV use the estimated PCV coverage among children ages <5 years during the study period was less than 10% countrywide and approximately 30% in central Thailand.

A rise of drug-resistant pneumococcus is a global concern [9-12,16].   A study of economic impact of infections attributable to drug-resistant organisms demonstrating that infections due to multidrug-resistant organisms are associated with higher mortality, longer hospital stays, and increased costs [18].  Currently, The Health Intervention and Technology Assessment Program (HITAP) in Thailand is conducting a cost-effectiveness study of introduction PCV into the NIP. Data regarding serotype distribution, serotype coverage of PCVs are crucial for this analysis for the country.  In 2021, a pilot project was launched of PCV-10 in a 2+1 schedule for children at 2, 4, and 12 months of age without catch up in one of the Northeastern provinces of Thailand. This underscores the importance of our current data reporting serotypes and antimicrobial susceptibilities of IPD prior to implementation of PCV in the NIP. “

Reference #18: Giraldi G, Montesano M, Napoli C, Frati P, La Russa R, Santurro A, Scopetti M, Orsi GB. Healthcare-Associated Infections Due to Multidrug-Resistant Organisms: a Surveillance Study on Extra Hospital Stay and Direct Costs. Curr Pharm Biotechnol. 2019;20(8):643-652. doi: 10.2174/1389201020666190408095811. PMID: 30961489.

Methods: 

Point 4: The survey was conducted using isolates from 2012-2016, I understand that the Covid-19 pandemic may have focused the researches on it, but the epidemiological data the author are presenting is old.

Response 4:

  • We would like to highlight that while the epidemiological period report was 2012-2016 there has been no other published data on serotypes and antimicrobial susceptibilities of IPD in Thailand since our prior publication which covered 2009-2012 [Hum Vaccin Immunother. 2014;10(7):1866-73]. We strongly believe that these data are important for health care professional in the region and should be published.

Point 5: The enrolment procedure must be specified. How did the authors choose the way to select the samples? This can represent a great bias origin. How did they avoid the selection bias? How did the authors choose the way to select the three hospital?

Response 5:

  • We included all culture positive samples from sterile specimens isolated from patients cared for in one of the 43 hospitals in our network. While selection bias is always a concern we believe there is minimal selective bias in our sample selection. We have added the enrollment procedure and described our clinical site network in the first paragraph of the materials and methods as follows:

“Pneumococcal isolates from sterile specimens of patients in a collaborative network of 43 hospitals in Central Thailand, which included Bangkok and 15 other central provinces. The study sites included 3 tertiary care University hospitals in Bangkok (Siriraj Hospital, King Chulalongkorn Memorial Hospital, and Bhumibol Adulyadej Hospital), and 40 public and private alliance hospitals. Samples collected between September 2012 and March 2016 were included in this analysis. We included all pneumococcal isolates from normal sterile sites that grew in culture from patients of all ages being cared for in one of the 43 hospitals in our network. Pneumococcal serotypes included in the PCV15 (serotype 4, 6B, 9V, 14, 18C, 19F, 23F, 1, 5, 7F, 3, 6A, 19A, 22F and 33F) were identified by Quellung reaction. Pneumococcal antisera from Statens Serum Institute (Copenhagen, Denmark) were used for serotyping. Calculation of serotype coverage was performed using serotypes within the vaccine, without considering potential serogroup cross-protection. Pneumococcal isolates that were not one of the PCV15 serotypes were defined as non-vaccine types (NVT). Each NVT was identified by sequential multiplex polymerase chain reaction (PCR) using the KAPA2G fast multiplex PCR kit (KAPA Biosystems Co., Ltd., USA), lysozyme (Sigma-Aldrich Co., Ltd., United Kingdom), tris-base (Affymetrix Co., Ltd., USA), EDTA (BIO BASIC Int., USA) and boric acid (Affymetrix Co., Ltd., USA) [19-20].”

Point 6: The authors do not propose a minimum sample size, what is the reference population? How large is it? Without the numerical identification of the reference population (all invasive infections due to S. pneumoniae) is not clear the validity of the study. A non-representative sample is by its self a non-sense-survey.

Response 6:

  • We did not propose a minimum sample size but included all positive culture samples (as described in our responses above). Data regarding the incidence of invasive pneumococcal disease (IPD) in Thailand and other countries in Southeast Asia are limited. Baggett, et al reported the first population-based estimates of the incidence of pneumococcal bacteremia in Southeast Asia. The annual incidence of hospitalized pneumococcal bacteremia in 2 provinces in Thailand was 3.7 and 7.6 cases per 100,000 persons in Sa Kaeo and Nakhon Phanom, respectively. In Thailand, antibiotics are available over the counter. Data from that study revealed that among 23,853 patients who had blood culture performed, 7620 (32%) had received antibiotics within 72 hours before collection of the culture specimen. The rate of antibiotic use among patients aged <5 years (33%) was similar to that among patients aged >5 years (32%). According to these incidence data, 276 pneumococcal isolates may represent a population of 3.6 to 7.4 million. We have added the small sample size as a limitation in the discussion as follows

“Our study had several limitations. Firstly, the majority of pneumococcal isolates collected were from patients in the Bangkok Metropolitan area. Although these data represent Central Thailand it may not necessarily represent the whole country. Secondly, we included 276 pneumococcal isolates in the study and while this may seem a relatively small sample size the first population-based estimates of the incidence of pneumococcal bacteremia in Southeast Asia by Baggett, et al (25) revealed that the annual incidence of hospitalized pneumococcal bacteremia in 2 provinces in Thailand was 3.7 and 7.6 cases per 100,000 persons in Sa Kaeo and Nakhon Phanom, respectively. In Thailand, antibiotics are available over the counter. Data from that study revealed that among 23,853 patients who had blood culture performed, 7620 (32%) had received antibiotics within 72 hours before collection of the culture specimen. The rate of antibiotic use among patients aged <5 years (33%) was similar to that among patients aged >5 years (32%). According to these incidence data, the 276 pneumococcal isolates in our study may represent a population of 3.6 to 7.4 million. Thirdly, this is a primarily a collaborative network of hospital laboratories to monitor the trend of serotype and antimicrobial susceptibility of IPD, therefore we have limited clinical data available. Finally, the period that we collected the samples was 2012-2016. Although this is now several years ago there has been no recent published data on serotypes and antimicrobial susceptibilities of IPD in Thailand since our last publication reporting between 2009-2012 (16). In 2021, a pilot project was launched of PCV-10 in a 2+1 schedule for children at 2, 4, and 12 months of age without catch up in one of the Northeastern provinces of Thailand. As such, this underscores the importance of our current data reporting serotypes and antimicrobial susceptibilities of IPD prior to implementation of PCV in the National Immunization Program. Furthermore, IPD is not a disease under surveillance by the Ministry of Public Health of Thailand.  Systemic Nationwide data collection, monitoring of serotype distribution and antimicrobial susceptibilities of pneumococcus causing IPD is crucial to guide treatment recommendation and future vaccine development.”

Point 7: Statistical analysis: I suggest to insert a measure of the magnitude of the effect for the comparisons. Please consider to include effect sizes.

Response 7:

  • We have added a table that include the odds ratio indicating magnitude of the effect for the comparison into Figure 4 as suggested.
  • We also added the sentence in Methods as following: “The odd ratio was analyzed to indicate effect size.”

Serotype

N (% Susceptibilities)

Penicillin

Cefotaxime

Meropenem

Erythromycin

Clindamycin

Co-trimoxazole

Ofloxacin

Tetracycline

Linezolid

Vancomycin

Non-19A

(n =240)

210 (87.5)

229 (95.4)

218 (90.8)

129 (53.8)

200 (83.3)

108 (45.0)

235 (97.9)

67 (27.9)

239 (99.6)

240 (100.0)

19A

(n =36)

27 (75.0)

32 (88.9)

19 (52.8)

12 (33.3)

34 (94.4)

6 (16.7)

36 (100.0)

8 (22.2)

36 (100.0)

36 (100.0)

p-value

0.045

0.116

<0.001

0.022

0.131

0.001

1

0.474

1

1

Odds ratio

2.336

2.604

8.850

2.326

0.294

4.098

-

1.355

-

-

Figure 4. Antimicrobial susceptibilities of pneumococcus causing IPD among serotype non-19A VS 19A in central Thailand, 2012-2016.

Discussion:

Point 8: the discussion need to be expanded. The Authors do not fully emphasize the contribution of the study to the literature.

Response 8:

  • We have added the contribution of this study in the discussion as follows

  1. Serotype distribution and serotype coverage by vaccines in the 2nd paragraph of the discussion:

“The serotype coverage rates of PCV10 in children <5 years and all ages were 55.8% and 53.3%, respectively. PCV13 provided similar coverage rates to that of PCV15, 71.3% and 72.1% in children <5 years and all ages, respectively. Thus, children should benefit from introduction of PCVs into Thailand’s NIP, with improved serotype coverage rates for children with PCV13 or investigational PCV15 compared to PCV10. According to a recent WHO position paper, both PCV10 and PCV13 have a substantial impact against pneumonia, vaccine-type IPD and NP carriage. Currently, there is insufficient evidence of a difference in the net impact of the 2 products on overall disease burden. PCV13 may have an additional benefit in settings where disease attributable to serotype 19A or serotype 6C is significant. The choice of product to be implemented should be based on programmatic characteristics, vaccine supply, vaccine price, the local and regional prevalence of vaccine serotypes and antimicrobial resistance patterns [28]. The serotype coverage rate of a PPSV23 for patients >2 years was 83.7%, approximately 30% and 10% higher compared to PCV10 and PCV13, respectively. Therefore, patients with high risk conditions of IPD in Thailand should benefit from immunization with PPSV23, in addition to PCVs.”

Reference #28. World Health Organization. Pneumococcal conjugate vaccines in infants and children under 5 years of age: WHO position paper. Wkly Epidemiol Rec 2019, 94, 85–104.

  1. Antimicrobial resistance

  • We emphasize the high rates of meropenem resistance of pneumococcus causing IPD in our network and to be cautious using meropenem as a first-line treatment.

“In Taiwan, ceftriaxone resistance according to non-meningitis criteria was identified in 38% of the IPD isolates from 2011 to 2013, and a major independent risk factor associated with inappropriate initial therapy that subsequently contributed to higher mortality. The majority (77.6%) of these isolates belonged to additional PCV13 serotypes, with approximately half being19A, 6A or 3 [35]. Therefore, the use of PCV13 in children as well as in the elderly population is likely to offer protection from the infection caused by antimicrobial-resistant pneumococci. Meropenem resistance of pneumococcus causing IPD in our network was 15.9%, and the majority were serotype 19A. Emerging meropenem resistance is expected to have a major impact on clinical practice in Thailand and in this context may not be the optional first-line treatment for patients. Of 629 IPD and non-IPD isolates from a nationwide surveillance in pediatric patients in Japan (2012 to 2014), the non-susceptible proportions to various antibiotics were: 46.1% to penicillin, 20.2% to cefotaxime, 20.5% to meropenem, 94.3% to erythromycin and 0.2% to levofloxacin. The non-susceptible rate to meropenem increased, particularly for serotype 15A [36]. High rates of resistance to meropenem is alarming.“

Point 9: The discussion is not updated in light of the economic impact of these infections (see the above mentioned reference).

Response 9:

  • We have added the economic impact of pneumococcal diseases in the Discussion.

Pneumococcal diseases significantly increase morbidity and mortality, with approximately half of these death occurring in children aged under 5 years. The consequences and deaths adversely impact individuals’ and caregivers’ work productivity. A study aimed to quantify the potential lifetime productivity loss due to pneumococcal diseases among the pediatric population in Thailand using productivity-adjusted life years (PALYs) revealed that 453,401 years of life and 457,598 PALYs would be lost to pneumococcal diseases, equating to a loss of US$5586 (95% CI 3338–10,302) million. Vaccination with PCV13 at birth was estimated to save 82,609 years of life and 93,759 PALYs, which equated to US$1144 (95% CI 367–2591) million in economic benefits. The disease and financial burden of pneumococcal diseases in Thailand is significant, but a large proportion of this is potentially preventable with vaccination [37]. These findings underscore the potential benefit of implementation of PCVs in to the NIP in Thailand and many developing countries.

Reference #37. Ounsirithupsakul, T.; Dilokthornsakul, P.; Kongpakwattana, K.; et al. Estimating the Productivity Burden of Pediatric Pneumococcal Disease in Thailand. Appl Health Econ Health Policy 2020, 18, 579-587.

Point 10: The Authors should add more practical recommendations for the reader, based on their findings.

Response 10:

  • We added information regarding choice of PCVs and recommended not to use meropenem as an initial treatment of suspected IPD due to high rates of resistance as mention above.

Point 11: Also, the section of limitations and future search is also very short, the Authors could elaborate on that.

Response 11:

  • We have added a paragraph highlighting the study limitations in the discussion as follows:

“Our study had several limitations. Firstly, the majority of pneumococcal isolates collected were from patients in the Bangkok Metropolitan area. Although these data represent Central Thailand it may not necessarily represent the whole country. Secondly, we included 276 pneumococcal isolates in the study and while this may seem a relatively small sample size the first population-based estimates of the incidence of pneumococcal bacteremia in Southeast Asia by Baggett, et al [25] revealed that the annual incidence of hospitalized pneumococcal bacteremia in was 3.7 and 7.6 cases per 100,000 persons in Sa Kaeo and Nakhon Phanom provinces in Thailand, respectively. According to these incidence data, the 276 pneumococcal isolates in our study may represent a population of 3.6 to 7.4 million. Thirdly, this is a primarily a collaborative network of hospital laboratories to monitor the trend of serotype and antimicrobial susceptibility of IPD, therefore we have limited clinical data available. Finally, the period that we collected the samples was 2012-2016. Although this is now several years ago there has been no recent published data on serotypes and antimicrobial susceptibilities of IPD in Thailand since our last publication reporting between 2009-2012 [16]. In 2021, a pilot project was launched of PCV-10 in a 2+1 schedule for children at 2, 4, and 12 months of age without catch up in one of the Northeastern provinces of Thailand. As such, this underscores the importance of our current data reporting serotypes and antimicrobial susceptibilities of IPD prior to implementation of PCV in the National Immunization Program. Furthermore, IPD is not a disease under surveillance by the Ministry of Public Health of Thailand.  Systemic Nationwide data collection, monitoring of serotype distribution and antimicrobial susceptibilities of pneumococcus causing IPD is crucial to guide treatment recommendation and future vaccine development.”

Round 4

Reviewer 1 Report

Dear Editor, I do apologize for my strong and repeated comments, but in my opinion they were necessary. After the revision process the paper was significantly improved and, in my opinion, is now suitable for publication.